# Vitamin E: A Review of Its Application and Methods of Detection When Combined with Implant Biomaterials

**DOI:** 10.3390/ma14133691

**Published:** 2021-07-01

**Authors:** Francesca Gamna, Silvia Spriano

**Affiliations:** Department of Applied Science and Technology, Politecnico di Torino, 10129 Torino, Italy; silvia.spriano@polito.it

**Keywords:** vitamin E, biomaterials, biomedical applications

## Abstract

Vitamin E is a common compound used for tocopherols and tocotrienols (α, β, γ, δ); it is the component of many natural products of both plant and animal origin. Thanks to its powerful antioxidant capacity, vitamin E has been very successful in hip and knee arthroplasty, used to confer resistance to oxidation to irradiated UHMWPE. The positive results of these studies have made vitamin E an important object of research in the biomedical field, highlighting other important properties, such as anti-bacterial, -inflammatory, and -cancer activities. In fact, there is an extensive literature dealing with vitamin E in different kinds of material processing, drug delivery, and development of surface coatings. Vitamin E is widely discussed in the literature, and it is possible to find many reviews that discuss the biological role of vitamin E and its applications in food packaging and cosmetics. However, to date, there is not a review that discusses the biomedical applications of vitamin E and that points to the methods used to detect it within a solid. This review specifically aims to compile research about new biomedical applications of vitamin E carried out in the last 20 years, with the intention of providing an overview of the methodologies used to combine it with implantable biomaterials, as well as to detect and characterize it within these materials.

## 1. Introduction

### 1.1. Structure of Vitamin E

Vitamin E was first discovered and described by Evans and Bishop in 1922; it includes eight natural forms (α, β, γ, δ; tocopherols and tocotrienols), and it can be found in various products bearing fats of both vegetable and animal origin, such as in olive or almond oil, hazelnuts, and egg yolk, and in the liver. Basically, tocopherols and tocotrienols have the same chemical structure, characterized by a 16-carbon lateral chain attached to position 2 of a benzopyran ring. The two isoforms differ substantially from the saturation of the long radical chain: tocopherols have a fully saturated chain, while tocotrienols have an unsaturated chain. As Figure 1 explains, the two homologs were named according to the position and number of the methyl group bound to the phenolic ring [1,2,3].

Among all the isoforms of vitamin E, α-tocopherol is the most abundant in the blood, because it is the only one that is absorbed within the body, while the other isoforms are excreted through the intestine. The liver, which takes up nutrients after they are absorbed by the small intestine, absorbs vitamin E thanks to the plasmatic lipoproteins that function as carriers. Among all the various forms of the vitamin E, alpha-tocopherol is re-secreted through the liver protein of α-tocopherol transfer (α-TTP) and distributed to circulating lipoproteins (LDL, IDL, VLDL). The other forms are metabolized and then expelled through the intestine (Figure 2) [4].

### 1.2. Biological Role of Vitamin E

Alpha-tocopherol is a phenolic antioxidant. The scavenging mechanism involves the donation of hydrogen from the hydroxyl group (-OH) of the phenolic ring to free radicals (ROS). In this way, the free radicals become unreactive and unable to do any more damage. After this reaction, also the phenolic compound itself becomes relatively unreactive with a higher stability. Its stability is guaranteed by the now unpaired electron which is on the oxygen atom and which is delocalized in the structure of the aromatic ring. α-tocopherol is located within the phospholipid membrane of the cell, and it occurs with the radical chain embedded in the hydrophobic core of the double layer [5]. Its concentration, compared to the lipids present in the membrane, is very low, but in spite of this, it plays an important role in preserving the integrity of the membrane by preventing lipid peroxidation which causes damage of cellular membranes, lipoproteins, and other molecules that contain lipids, in conditions of oxidative stress [1,6,7].

Oxidative stress is a pathological condition caused by the imbalance between the generation and elimination of chemical oxidant species (ROS), and it is involved in several neurodegenerative diseases such as Alzheimer’s and Parkinson’s disease that are implicated in free radical processes and oxidative damage [8]. That said, it is easy to think of vitamin E that, thanks to its important qualities as an antioxidant, may have an important role in the integrity of the brain. To confirm this, a high level of α-TTP was found in the brain [9].

Vitamin E is an important anti-inflammatory molecule since it acts on many different factors that affect, directly or indirectly, the immune system. Vitamin E is able to modulate inflammation through different ways: it has effect on proinflammatory enzymes such as COX, responsible for prostaglandins (PG)E_2_ production [10,11]; (PG)E2 is a proinflammatory mediator that has been associated with several senility-related diseases such as cancer, arthritis, and cardiovascular diseases [4,7,12]). It modulates the proliferation and activation of certain cells of the immune system such as T cells, lymphocytes, and NK cells [13]. Finally, it acts on the secretion of proinflammatory cytokines such as IL-6 and TNF-α. Thanks to these factors, vitamin E plays an important role in helping to prevent chronic inflammation [11,13]. Chronic inflammation is strictly linked to oxidative stress [14] and, together with it, it is the main cause of age-related disease and cancer [11].

Vitamin E has also an important role in the inhibition of platelet aggregation, inhibiting various enzymes such as PKC, which is a key signal transduction pathway in several cell types [1,15]. α-tocopherol, thanks to its free-radical scavenging and anti-inflammatory properties, has a benefit also in dermatology: it protects the skin from UV radiation, and accelerates the wound healing process after an injury such as ulcers or burns [16]. Inflammation could be associated also to a large number of different phenomena related to bone health: it is thought that thanks to its anti-inflammatory action and regulation of cytokine secretion, vitamin E can influence bone remodelling, being able to protect the bone against osteoclastic activity, increasing osteoblasts differentiation, and protecting cartilage health [17,18,19]. In addition, recent studies have shown that vitamin E also has biomechanical effects, being able to increase specific characteristics such as load and yield [20]. Together with all these considerations, vitamin E, due to its antioxidant function, role in anti-inflammatory processes, inhibition of platelet aggregation, and immune system-enhancing activity [1,9], brings a wide range of benefits, from anti-cancer effects [21] to the prevention of disease progression, and in improving quality of life in the elderly [22]. The figure below (Figure 3) shows a schematic representation of the biological role of vitamin E.

More controversial and still under study, is the anti-bacterial role of vitamin E. In the literature, it is possible to find vitamin E used as a natural compound to treat infections caused by specific Gram-positive or negative bacteria [23,24] or as an antibiotic adjuvant used in combination with antibiotics for the treatment of infections [25,26]. Moreover, the addition of vitamin E to materials may have the ability to reduce biofilm formation on the surface. As the literature suggests, it is possible to believe that tocopherol is able to reduce the biofilm formation capacity of a big range of strains (S. aureus and S. epidermidis, etc.), regardless of the classification of the bacterium (Gram-negative or -positive) [27]. This, however, comes into opposition with other studies that show instead that vitamin E, in composition with other materials, did not reduce biofilm formation [28].

Once the biological role of vitamin E is understood, it is certainly interesting to analyse its application and detection methods, when it is combined to a biomaterial. Over the past 20 years, there has been a strong increase in interest in scientific research on α-tocopherol combined with various biomaterials (Figure 4). This review selected about 100 reviews best suited to the topic discussed.

## 2. Biomedical Applications of Vitamin E

### 2.1. Vitamin E in Prosthetic Implants

Vitamin E is being researched as a molecule incorporated or used as a coating on prosthetic materials with the aim of creating implants with additional properties.

#### 2.1.1. Vitamin E in Blend Form or through Bulk Diffusion

So far, the most successful application of α-tocopherol has been the use as an antioxidant to stabilize high molecular weight polyethylene (UHWMPE), which has made vitamin E a useful compound in the biomedical field. Until the late 1990s, prior to implantation, UHWMPE inserts were sterilized with gamma rays. Unfortunately, this type of sterilization carried a major oxidative problem for the polymer, since the radiation increases the possibility of oxidative degradation, making it more unstable and fragile, leading to severe implant loosening phenomena [25,27]. The incorporation into the polymer of antioxidants such as tocopherols was considered to overcome this problem. This idea was born from the use of vitamin E in food packaging industries as an antioxidant of polyolefins [29,30]. For the incorporation phase, α-tocopherol is typically inserted into the material before sterilization in two ways, before (creating the VE-UHMWPE blend) or after crosslinking irradiation (through bulk diffusion) [31,32,33]. The success of this new material in the biomedical field is guaranteed by the very good biocompatibility of vitamin E which has been proven not to have cytotoxic effects [25]. After the use of alpha-tocopherol combined with UHWMPE, new combinations with other biomaterials have been pursued and are still under study. 

#### 2.1.2. Vitamin E as a Coating

One of the most common concerns of biomedical implants is the problem of joint replacement because of loosening. Causes of failure include infections and wear, which could cause chronic inflammation by the presence of debris originating in the articular part of the prosthesis. A coating able to have an anti-inflammatory and anti-bacterial action can certainly help to reduce these events, consequently reducing the phenomena of septic or aseptic loosening of the prostheses. For this reason, although still in the research phase, some works suggest the use of vitamin E for coating metal surfaces with the aim of providing anti-bacterial and anti-inflammatory properties or to help osseointegration [34,35,36,37]. These studies employed pure titanium as a substrate, which is a biomaterial commonly used in dental, cranial, and joint implants. As later shown in Table 1, the coatings are created by the adsorption of vitamin E on the metal surfaces by leaving the samples in the vitamin solution. Given the high hydrophobicity of vitamin E, in all cases the solvent used during the coating formation is ethanol. Then, the biological response (in vitro and in vivo) of the titanium implants coated with vitamin E has been evaluated. Bidossi et al. made interesting studies on the anti-microbial activity, creating coatings on pure titanium surfaces, using two different forms of vitamin E: α-tocopherol acetate and α-tocopherol phosphate, derived from the esterification of α-tocopherol with the acetate group and the phosphate group, respectively. It was found that especially tocopherol phosphate is anti-microbial against both Gram-positive and Gram-negative, and it also has the ability to stimulate osteoblasts, finding many applications in the implant field [31,34]. Maria Satuè et al. used vitamin E as an antioxidant, recalling its main ability that protects the 7-DHC molecule; this molecule during UV irradiation is activated, becoming a precursor of vitamin D, and it is employed as a coating on pure titanium surfaces [33,34].

### 2.2. Vitamin E in Tissue Engineering

Thanks to its antioxidant and anti-inflammatory ability, α-tocopherol has become an interesting object of research in the world of wound healing and tissue regeneration. In the literature, vitamin E can be found combined, encapsulated, or blended with different types of polymeric materials treated either as scaffolds, hydrogels, or films.

#### 2.2.1. Vitamin E in Wound Healing

Wound healing is a physiological process to maintain and achieve skin integrity after an injury such as a laceration or burn. Wound healing is a complicated and time-consuming process, as it consists of the following steps: inflammation, proliferation, and maturation [38,39]. Combining vitamin E with polymers typically used in biomedicine, it is possible to obtain polymeric wound dressing that harnesses the antioxidant capabilities of vitamin E, so as to help and accelerate the healing process. For complete skin regeneration, the preferred dressing media are hydrogels and scaffolds, as they are three-dimensional and facilitate cell attachment and growth. In the biomedical field, from tissue engineering to drug delivery, biopolymers commonly used are biodegradable polyesters, especially poly-e-caprolactone (PCL), poly(lactide) (PLA), and poly(lactide-co-glicolide) (PLGA). These materials ensure biocompatibility, biodegradability, and good mechanical performances. Polysaccharides, including chitosan, hyaluronic acid, and cellulose, in addition to having good biocompatibility and many hydrolysable groups, also have a structure that mimics the extracellular matrix (ECM). Chitosan hydrogel is an example of a perfect biomaterial for wound dressing; in fact, Arian Ehterami et al. prepared and tested chitosan/alginate hydrogel loaded with vitamin E for dorsal skin wound healing in a rat model that ensured epidermal cell proliferation with even a new generation of hair follicles, promising successful wound healing [40]. As mentioned, in addition to hydrogels, scaffolds are widely used for this purpose. However, the use of polyesters, typically with better mechanical performance than polysaccharides, may not provide a favourable surface for cell attachment. In fact, Chinnasamy Gandhimathi et al. produced PLA-CL nanofiber scaffolds, through electrospinning, and incorporated silk fibroin to enhance cell adhesion and vitamin E to provide a low-stress environment for cells. The created scaffolds were able to induce fibroblast proliferation and collagen secretion [41]. Saba Zahid et al. instead thought of using the two biomaterials, but for obtaining two layers, one of porous PCL and the other one of PLA film incorporating vitamin E; in this way, they were able to obtain a bilayer of electrospun nanofibers that supports cell proliferation and also angiogenesis [42]. On the other hand, polysaccharide scaffolds do not present the above problem: electrospun cellulose acetate mats loaded with vitamin E prove effective for dermal therapy due to greater flexibility and good vitamin E release kinetics [43].

In addition to three-dimensional structures, polymeric films can be more easily obtained to aid wound healing. Gabriela Garrastazu Pereira et al. selected two polysaccharides, hyaluronic acid, and sodium alginate to create a polymeric film loaded with vitamin E acetate and aloe vera, characterizing it both mechanically and chemically [44]. Similarly, Sonia Trombino et al. prepared a collagen film esterified with α-tocopherol (collagen α-Tocopherulate) for wound-healing applications, exploiting the antioxidant capacity of the molecule [45].

#### 2.2.2. Vitamin E in Tissue Regeneration

Tissue repair refers to the compensatory regeneration of a tissue, resulting in restoration of tissue structure and function. Tissues can be repaired by implanting structures, preferably three-dimensional, such as scaffolds that are able to stimulate cell proliferation and adhesion for complete tissue regeneration [46]. In this context, the same materials as above are used, which can support the tissue both mechanically and at the cellular level, and after doing their job are able to degrade in a physiological environment. Generally, polymers with good mechanical properties are used for hard tissues, while for soft tissues such as the cardiac one, a high mechanical performance is not required.

For example, Poly(3-hydroxybutyrate) (P(3HB)) and bioactive glass are shown to have favourable properties for hard tissue regeneration. Superb K. Misra et al. created a Poly(3-hydroxybutyrate) (P(3HB))/bioactive glass scaffold, adding vitamin E as an antioxidant capable to enhance protein adsorption [47]. Instead, contrary to what they expected, Filippo Renò et al. prepared PLA films blended with vitamin E which induce a reduction in osteoblast cell attachment and spreading [48]. This research team has published many works on these films. Their studies reveal that vitamin E confers important characteristics to the PLA surface: anti-adhesion for bacteria, more hydrophilic and therefore higher protein adsorption, and anti-adhesion for osteblasts, obtaining a new type of polymer that can be used in tissue engineering with several biological effects [48,49,50]. Besides a biological role, vitamin E also has a role in sustaining and adhering two different materials: Iulian Antoniac et al. created PLA-Mg composites as a filament for biomedical applications and used vitamin E to enhance the adhesion between magnesium and PLA [51]. Instead, Zahra Mahdieh et al. employed vitamin E to protect starch, a biodegradable polysaccharide with low mechanical performance, from oxidation during the blending process at an elevated temperature [52].

For soft tissues, Youyang Qu et al. prepared alpha-tocopherol liposome loaded in chitosan hydrogel. α-tocopherol was entrapped in a liposomal carrier, and the liposome was formulated into a chitosan-based hydrogel, with the aim of creating injectable engineered cardiac tissue capable of suppressing oxidative stress in the microenvironment, supporting and surviving cardiomyocytes [53]. Due to the hydrophobic nature of vitamin E, it is always a bit of a struggle to insert it into polymers that are typically hydrophilic because they need to be hydrolysable. Table 2 reports in more detail methods of loading vitamin E in different biomaterials for tissue engineering applications and each biological response. As it can be seen, vitamin E can be inserted within the dressing through two ways: inserted directly within the solution, creating a homogenous mixture, which is then electrospun, gelled, or polymerized, or inserted within nanocarriers that are then in turn inserted within the biomaterial.

### 2.3. Vitamin E in Drug Delivery

#### 2.3.1. Pharmaceutical Use of Vitamin E

Nowadays, vitamin E has also gained particular importance due to its proven anti-carcinogenic activities, which lead it to be an interesting candidate as an adjuvant anti-cancer treatment drug or as a preventive drug. In fact, its preventive properties were discovered according to research that showed that the Mediterranean diet, known to be rich in antioxidants such as vitamin C and E, has a protective effect from certain cancers, such as colon cancer [21,54]. As the literature suggests, γ- and δ-tocopherol are more potent inducers of apoptosis than β- and α-tocopherols; in fact, despite its antioxidant power, it would seem that α-tocopherol is not cytotoxic [55]. In particular, among all the various forms of vitamin E, tocotrienols appear to have a stronger anti-proliferative and proapoptotic effect than tocopherols [56]. Due to its lipophilicity, vitamin E is able to cross the cell membrane but, given its insolubility in water, its bioavailability is limited. In fact, the biological mechanism of digestion of vitamin E occurs through emulsion into lipid droplets. The emulsion allows vitamin E to be transported (through micelles or vesicles) and thus to be absorbed by diffusion by the various target tissues [57,58]. For this reason, methods to encapsulate vitamin E for the use in drug delivery have been studied in the literature, using various materials such as synthetic polymers or biopolymers and different delivery systems: hydrogel, micro- and nano-particles, liposomes, and nano-emulsions.

In the literature, chitosan is widely explored as a drug delivery system to load alpha-tocopherol: the water solubility of chitosan is exploited to increase bioavailability of α-tocopherol.

Majid Naghibzadeh et al. developed chitosan nanoparticles with alpha-tocopherol loaded by dispersing chitosan in water and dropping a solution of ethanol and α-tocopherol. The nanoparticles are obtained by placing the mixtures in an ice bath and sonicated using the ultrasonic probe [59]. With a similar approach, J. Nam et al. created a chitosan-specific micelle for tocopherol and doxorubicin (DOX) delivery by grafting with a targeting ligand (anti-HER2/neu-polyethylene glycol [PEG] peptide). In this way, micelles have an anti-cancer effect, exploiting the synergistic effect between TP and DOX, and with site-specific drug delivery [60]. So, chitosan is a good biomaterial that can encapsulate tocopherol by O/W emulsion either on its own or with the help of other polymers such as zein, an amphiphilic protein found in corn, which helps create a stronger polymeric complex with chitosan. In fact, Yangchao Luo et al. prepared and characterized nanoparticles by first preparing a solution in ethanol of zein and tocopherol and then, inserting into this another solution of chitosan and acetic acid [61].

In addition to chitosan, PCL was also investigated for drug delivery systems. Similar to that mentioned about chitosan, PCL nanoparticles loaded with tocopherol were made into O/W emulsion with a successive ultrasonification method to optimize encapsulation. Youngjae Byun et al. formulate and characterize PCL nanoparticles loaded with tocopherol, using PCL dissolved in methylene chloride with tocopherol as an organic phase and PVA dissolved in PBS as an aqueous phase, with a subsequent ultrasonification in an ice bath [62]. Instead, Catherine Charcosset et al. created a PCL nanocapsule, using PCL dissolved in acetone with tocopherol as an organic phase and TWEEN 20 dissolved in water as an aqueous phase [63].

Another important biopolymer used in drug delivery is definitely hyaluronic acid, a hydrophilic compound, capable of forming a gel when immersed in water, which could be used also to enhance the water solubility of alpha-tocopherol. Parbeen Singh et al. created hyaluronic-acid-based β-cyclodextrin grafted copolymer to encapsulate vitamin E [64].

Moving on to another area, temperature-sensitive materials have received a great deal of attention in the drug delivery field, as they can optimize the encapsulation and controlled release of drugs. Cirley Quintero et al. used a PNIPAM-b-PCL-b-PNIPAM triblock copolymer as a thermosensitive material for nanoparticles. The obtained nanoparticles have a shell of the aforementioned polymer, loaded with α-tocopherol. Their preparation is based on dissolution of α-tocopherol and the corresponding copolymer in acetone, and a subsequent drop in an aqueous solution at pH 5 [65]. Likewise, Behrouz Mohammadi et al. have successfully nano-encapsulated vitamin E in stearic acid-lauric acid (SA-LA) in the form of a core-shell thermosensitive structure. The synthesis process involves always the O/W emulsion with SA-LA and α-tocopherol as an organic phase and PVA, PVP, Sodium SLS as an aqueous phase [66].

From what has been said so far, we can see that in the literature, typically vitamin E is encapsulated within different polymeric vectors (Table 3). There are also cases where α-tocopherol is functionalized on metal nanoparticles such as gold to increase the scavenging ability of the latter for biological applications [67,68].

#### 2.3.2. Vitamin E as a Delivery System

Most of the approved anti-cancer drugs, such as paclitaxel, docetaxel, etc., are lipophilic, which renders it challenging for their bioavailability and thus for their uptake by the target tissues [69].

Although tocopherols and tocotrienols have a nearly amphipathic structure, the hydrophilic part of the tocols, characterized by the hydroxyl group (-OH), is too small to self-assemble spontaneously into a micellar structure. Conversion of the phenolic component of the vitamin to esters using acetic or succinic acid is often performed to expand the hydrophilic part and optimize the amphiphilic structure, thus creating more stable esterified compounds that can be easily used for drug delivery, such as α-tocopheryl ether-linked acetic acid (α-TEA) or α-tocopherol succinate (α-TOS), also discovered to be potent anti-cancer agents [55,70,71]. The esterification of α-tocopherol succinate (α-TOS) with the important biomaterial, polyethylene glycol (PEG), forms a new compound, D-a-tocopheryl polyethylene glycol succinate (TPGS), widely used today in drug delivery as a non-ionic surfactant and micellar stabilizer, capable of forming micelles in water at a concentration as low as 0.02 wt%. This new amphiphilic polymer has several advantages given by the physicochemical properties of PEG and vitamin E, such as high biocompatibility, improved cellular uptake of the drug, and anti-tumour activity, which allowed it to be approved by the FDA as a safe adjuvant [61,62,63]. TPGS is widely used as a surfactant and permeation enhancer (reviewed in [61]), as a prodrug carrier (reviewed in [72,73]), and also as a copolymer with other biopolymers such as PLA and PLGA or PCL (reviewed in [74,75]).

Another interesting synthetic polymer is formed by the esterification of alpha-tocopherol succinate with the glucidic polymer, inulin (INU), creating amphiphilic compounds obtained based on inulin with hydrolyzable groups, and with a vitamin E-based radical chain, called INVITE. This novel INU-based polymer easily self-assembles into nanocarriers; thus, it has been envisioned as a novel drug delivery system for the therapy of different diseases.

Delia Mandracchia et al. developed and characterized INVITE micelles, which have stability in water and a low size (about 50 nm), and is used in different diseases for different targets. Due to the renal passive targeting ability of inulin, INVITE results as a good candidate for the drug delivery system for target tissues such as the urinary tract [76]. It has also been tested with good results for the delivery of celecoxib to the intestinal site against Caco-2 cells [77] conjugated with several other molecules such as biotin and succinic anhydride, creating INVITEBIO and INVITESA, respectively [78,79], and loaded with curcumin, creating INVITE C, for the treatment of diabetic retinopathy or neurodegenerative diseases [80,81]. Through the esterification of alpha-tocopherol succinate with hyaluronic acid, a novel amphiphilic compound was created for the tumour-targeted delivery system, exploiting the ability of hyaluronic acid to bind to the membrane receptor CD44, a protein overexpressed by tumour cells. The created polymer name is HA-VES and was used by Jinling Wang et al. for the release of anti-cancer agent Doxorubicin [82].

That said, as can be seen from Figure 5, vitamin E is often bound with other biomolecules able to make the molecule more hydrophilic.

This creates amphiphilic compounds that are able to reorganize themselves into systems suitable for drug delivery. In contrast, Khushwinder kaur et al. have succeeded in creating nanoparticles of pure α-tocopherol. The work suggests a method that involves an emulsion of the components with a water:surfactant:oil ratio where the organic phase consists of the first surfactant TWEEN 80 and tocopherol in ethanol, and the aqueous phase is water containing the second surfactant SSL. Through this method, they were able to create stable nanoparticles for the encapsulation of curcumin as an antioxidant and benzylisothiocyanate as an anti-cancer agent [83].

## 3. Methods to Detect and Quantify Vitamin E within Materials

Vitamin E can be easily detected in different liquid media, such as in oils, serum, human milk, foods, etc., by different methods including HPLC, FTIR, RAMAN, UV-VIS, and spectrophotometric methods. Recently, some protocols were also developed for the analysis of vitamin E incorporated into cosmetics and food packaging and contained in food [84,85,86,87,88,89].

The first method usually used to allow a simple and rapid quantitative determination of α-tocopherol is High-Performance Liquid Chromatography (HPLC). In fact, HPLC is one of the most powerful tools for the determination of fat-soluble vitamins and has been widely used for their separation and detection; different detectors can be used for vitamins such as UV-VIS, fluorescence, and mass spectrometry. In the case of vitamin E, typically the HPLC column is connected to an UV absorbance detector as the compound absorbs the ultraviolet light, particularly around 290 nm. This method is, in fact, used not only to analyse quantitatively the content of alpha-tocopherol in food or beverages, but also in cosmetics and in biological samples including human plasma and human milk [86,89,90,91,92]. Another way to detect and quantify α-tocopherol is the Fourier Transform Infrared Spectroscopy (FTIR). Sandra et al. developed a quick procedure for the quantitative analysis of α-tocopherol in vegetable oils as an alternative to HPLC methods, using FTIR-ATR methodology. By analysing 13 vegetable oils, with a known content of vitamin E, a research team created a calibration curve which was then used to measure the alpha-tocopherol content of the vegetable oil concerned quantitatively [93].

For qualitative measurements, FTIR was also used for detecting the functional groups of a hydrophobic film of vitamin E deposited on a copper substrate [94]. Thanks to its clear absorbance peak at 290 nm, visible ultraviolet spectroscopy (UV-Vis) proved to be able to detect the presence of vitamin E even at low concentrations [95,96].

Along with FTIR, RAMAN is a potential alternative method to have qualitative detection of the molecule. It is used to detect vitamin E in oil water emulsions and in biological samples [97,98]. Surface-enhanced Raman spectroscopy (SERS) technology is of a high level of interest: it exploits the amplification of Raman diffusion by molecules adsorbed on a metal or on metallic nanoparticles [99,100]. Typically, most SERS techniques use metal aqueous colloids as a substrate, which require that compounds to be analysed must be water soluble. For water insoluble analytes, such as vitamin E, the matter is more complicated. Given the disadvantage of using colloidal Ag nanoparticles to measure SERS of the analyte directly, Tiantian Cai et al. have successfully tried other methods to analyse vitamin E: after dissolving the compound in chloroform, the solution obtained is dripped onto the surface of a metal substrate with surface Raman activity. Another method could be to immerse the metal substrate in the sample solution containing vitamin E directly, to extract it after a certain time, and to measure it at RAMAN after the solvent has evaporated [101].

Thanks to its antioxidant properties, vitamin E can also be analysed and quantified through all those methods that exploit chemical reactions, typically redox, to develop coloured compounds that are then measured spectrophotometrically. In general, spectrophotometric methods for vitamin E determination use oxidation of the aromatic ring of α-tocopherol, creating tocopherylquinone, by oxidizing agents that ultimately yield products with spectrophotometric staining. Among these methods, there is definitely the DPPH method, which uses a free radical of purple colour, which discolours when it reacts with vitamin E. Valeria M. et al. have used the DPPH method to compare the antioxidant power of drugs containing alpha-tocopherol. The problem of the DPPH method is its low reproducibility due to the low stability of the radical [102].

Another such methodology is the Folin–Ciocâlteu (FC) reagent in an aqueous solution. In this case, however, given the insolubility of vitamin E in water, this methodology is not the optimal one. However, modifications have been made to the FC method to enable the measurement of lipophilic and hydrophilic antioxidants concentrations simultaneously [103].

Albeit less used and dated, there are many other methods using different oxidizing reagents such as Fe(III)-bathophenanthroline, Cu(II)-neocuproine, or silver nitrate, but they require a rigid control of the conditions for precise results [104]. Another method is the Emmerie and Engel colour reaction with ferric chloride: it is a precise and easy-to-perform reaction, and therefore the approach of choice for a routine clinical laboratory [105]. Based on this work, more recently, Jameel G. Jargar et al. took advantage of different reagents such as 2,2′-bipyridyl, ferric chloride, and xylene to perform the colour reaction [106].

Finally, although a very old and no longer used method, nitric acid combined with ethanol was used to oxidise α-tocopherol, forming the coloured red o-quinone which can be detected spectrophotometrically (Figure 6) [107].

Thanks to the vitamin E detection methods employed in various applications involving cosmetics, food packaging, and so on, it is possible to apply the above methods to the biomedical field for the detection of vitamin E when combined with different biomaterials.

Certainly, it is easy to find detection methods in the literature when vitamin E is combined with UHWMPE, as it is the most widely applied biomaterial coupled with vitamin E today.

For quantification methods, with HPLC analysis, it is convenient to quantify the vitamin E content within UHMWPE using a calibration curve produced from the areas of the HPLC peaks [108]. Instead, Hufen Julia et al. developed an accurate method to detect α-tocopherol content in UHWMPE using HPLC analysis to separate it and determine its concentration by UV-Vis spectroscopy with a corresponding calibration curve [109]. However, it is also possible to use only UV-VIS in absorbance mode combined with FTIR to quantify vitamin E within UHMWPE [110]. Vitamin E blended with polyethylene induces yellowing of the sample; Martínez-Morlanes et al. exploited this characteristic using the colorimetric technique and reflectance spectroscopy to detect vitamin E embedded in polyethylene samples quantitatively [111]. These types of methods, especially HPLC, are also used in drug delivery to calculate the drug encapsulation efficiency, resulting in the quantification of the vitamin E encapsulated within polymeric nanoparticles [62,64,65]. With the same object, in tissue engineering, HPLC is used to quantify vitamin E content inside the matrices or scaffolds [44,112].

For qualitative methods, since vitamin E is an extremely hydrophobic molecule, another important way to detect the presence of tocopherol on different substrates is definitely the measurement of the contact angle, the quickest test to evaluate a surface modification [113]. Filippo Renò et al. used the contact angle measurement on PLA blended with Vitamin E, and they discovered that the blend enriched with vitamin E was more easily wetted [48,49].

To get a more in-depth understanding of the chemical bonds between the substrate and the deposited molecule, the XPS technique is useful, as in the case of Elena Stoleru et al. who used XPS to have information about the stability of the chitosan/vitamin E coating deposited on a polyethylene substrate [113].

Once the characteristic peaks of vitamin E are known, the FTIR analysis is helpful, not only to detect vitamin E [42], but also to analyse the eventual shifts in wavenumber of the peaks that denote an interaction between tocopherol and the combined biomaterials. Ahmad Salawi et al. used the FTIR technique to analyse the interaction between a new copolymer called Soluplus and α-tocopherol for a wound-healing application [114], and Joana T. Martins et al. studied the physiochemical effect of the incorporation of α-tocopherol in chitosan-based films through a different analysis including FTIR [115].

In the biomedical field, the DPPH test is used to analyse the radical scavenging ability of vitamin E combined with biomaterials, as Elena Stoleru et al. did on a film electrosprayed with a chitosan/vitamin E formulation [113]. DPPH was used also by Zhou Nier et al. to test the antioxidant activity of Au nanoparticles functionalized with Trolox (hydrophilic analogue of alpha-tocopherol) [67,68]. The table (Table 4) reports the characterization methods used to detect vitamin E when combined with different biomaterials.

## 4. Conclusions

This review is intended to give a general overview over the past 20 years of the biomedical use of vitamin E in all its forms. It sets out to recall its biological role, tracing the fundamental motives of why the molecule has become a widely discussed object in scientific research. Vitamin E has a number of relevant biological actions (antioxidant properties, anti-inflammatory properties, inhibition of platelet aggregation, preservation of integrity of the cell membrane) with a positive outcome on immune system, bone health, protection of the skin, age-related diseases, and cancer.

Apart from dietary supplements or food additives, several biomedical products on the market include vitamin E and exploit its benefits (wound healing, UHMWPE inserts); other current commercial applications are in cosmetics and food packaging. Some new fields of application are investigated in the area of biomaterials at the research stage and are of a great potential impact in the future; in this respect, we can mention coatings of bone implants, drug delivery systems, and tissue (both hard and soft) regeneration.

Anti-microbial action of vitamin E and its derivate compounds is still under discussion.

In the perspective of the future, further research must be carried out to understand the controversial antibacterial activity of vitamin E better, in order to be able to give new properties to other biomedical materials, in addition to those already discussed. New types of combinations with biomaterials still need to be studied and optimized, such as the grafting of the molecule on metal surfaces through coating or functionalization.

## Figures and Tables

**Figure 1 materials-14-03691-f001:**
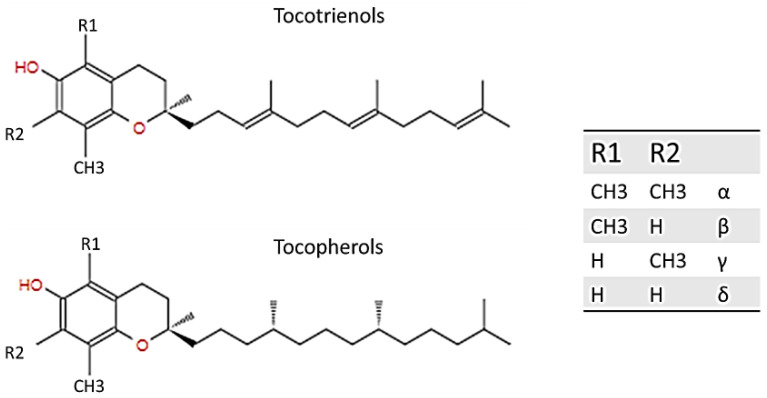
Structure of tocotrienol and tocopherol.

**Figure 2 materials-14-03691-f002:**
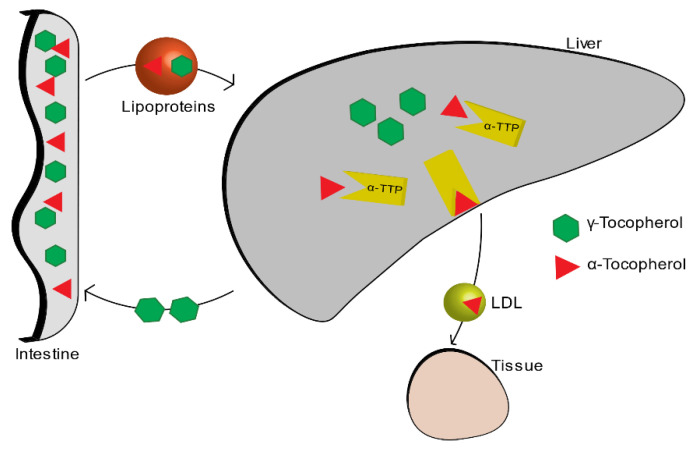
Vitamin E metabolism.

**Figure 3 materials-14-03691-f003:**
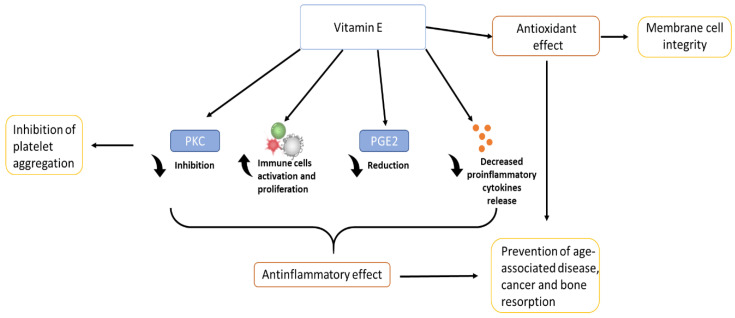
Biological role of vitamin E.

**Figure 4 materials-14-03691-f004:**
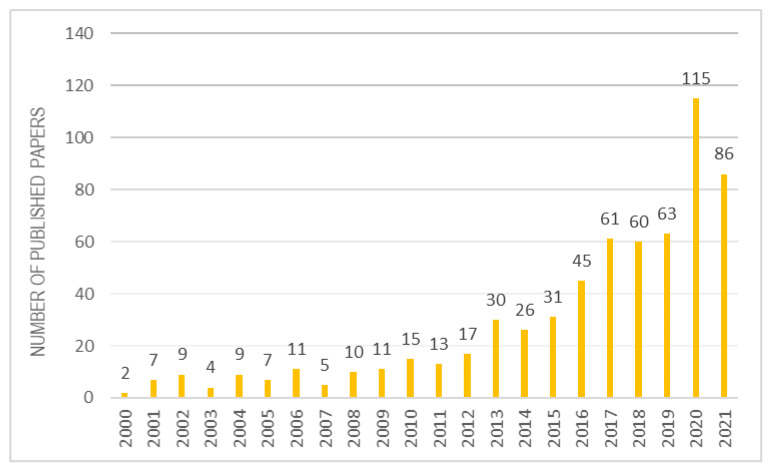
Studies inherent to α-tocopherol combined with biomaterials over the years.

**Figure 5 materials-14-03691-f005:**
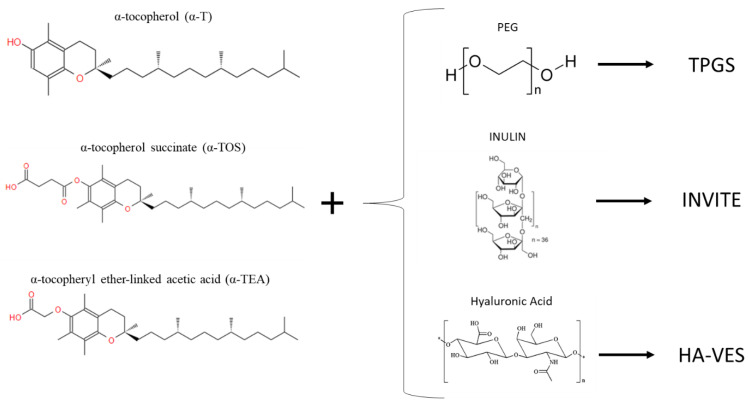
Chemical structure of different forms of vitamin E used for drug delivery system.

**Figure 6 materials-14-03691-f006:**
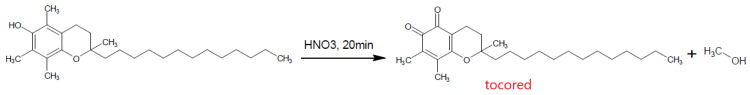
Formation of Tocored with Nitric Acid.

**Table 1 materials-14-03691-t001:** Vitamin E in Prosthetic devices.

Material	Molecule	Combination	Method of Combination	Results	Application	Ref.
**UHWMPE**	α-Toc	Blending	1: Blending of vitamin E with UHMWPE resin powder and following irradiation for crosslinking	Vitamin E is able to prevent oxidation during gamma sterilization of UHWMPE.	Material stabilizer	[31,32,33]
			2: Vitamin E is diffused into an already crosslinked UHWMPE			
**Pure Titanium**	α-toc + 7-DHC	Coating	Preparation of a solution of 7-DHC + tocopherol (1:1) in ethanol 10 µL of the prepared solution left on the surface and further UV-irradiated and incubate for 48 h at 23 °C	Vitamin E is able to protect 7-DHC from oxidation during UV-irradiation; vitamin E helps osseointegration of the coated samples both in vivo and in vitro.	Molecule stabilizer	[35,36]
	α-toc Acetate	Coating	Spreading of the solution of α-toc Acetate in ethanol (500 mg/mL) on a sandblasted disk of titanium.	Low anti-microbial activity shown in vitro in part against *S. aureus* and *S. epidermidis*.	Coating for implants in order to prevent implant-associated infections	[34]
	α-toc Phosphate	Coating	Spreading of a solution of α-toc Acetate in ethanol (500 mg/mL) on a sandblasted disk of titanium.	Good anti-microbial activity shown in vitro against every strain (*S. aureus* and *S. epidermidis*, *P. aeruginosa*, *P. acnes*). Good bone stimulation shown in vivo.	Coatings for dental implant in order to prevent implant-associated infections and help osseointegration	[34,37]

**Table 2 materials-14-03691-t002:** Vitamin E in Tissue Engineering.

Polymer Class	Polymer Name	Molecule	Structure	Method Loading	Biological Response	Role of Vitamin E	Applications	Ref.
**Polyesters**	PLA-Mg	α-toc	Filaments	Addition of Vitamin E in a PLA-Mg solution before the extrusion process and following 3D printing of the implant	Not reported	It enhances adhesion between Mg particles and PLA	Implant devices	[51]
	PLA	α-toc	Polymeric film	Addition of Vitamin E into the PLA/ chlorophorm solution. Shaking of the solution with following addition of it in glass dishes. Evaporation of the solvent in dark conditions at room temperature.	The film has good protein adsorption; inhibits osteoblasts, bacterial, and platelets adhesion and spreading.	Protein adsorption Antioxidant	Tissue engineering	[48,49,50]
	PLA-CL	α-toc, Silk Fibroin (SF), Curcumin (C)	Nanofiber scaffold	Preparation of a solution with PLACL, SF, C, and α-Toc in HFIP. Electrospinning of the prepared solution.	The scaffold induces fibroblasts proliferation and attachment, and stimulates collagen secretion.	Antioxidant and anti-inflammatory, wound-healing capacity	Wound treatment	[41]
	PLA and PCL	α-toc acetate	Bilayer nanofiber scaffold	Preparation of a solution with PCL, PLA, and Vitamin E in DCM. Electrospinning of the prepared solution.	The scaffold induces angiogenesis and cell proliferation	Antioxidant and anti-inflammatory, wound-healing capacity	Wound treatment	[42]
	P(3HB)/Bioglass	Vitamin E	Foam scaffold	Incorporation of the appropriate amounts of vitamin E and MWCNTs into the polymer solution. Sonication of the mixture before impregnating it into the preforms.	The scaffold has good biocompatibility, good protein adsorption; stimulates cell proliferation and allows vascularization.	Protein adsorption Antioxidant	Bone tissue engineering	[47]

**Table 3 materials-14-03691-t003:** Polymeric vectors for Vitamin E encapsulation.

Polymer	Molecule	Type of Vector	Ref.
Chitosan	α-toc	Nanoparticles	[59]
Chitosan	α-toc and Doxorubicin	Micelles	[60]
Zein	α-toc	Nanoparticles	[61]
PCL	α-toc	Nanoparticles	[62]
PCL	α-toc	Nanocapsule	[63]
Hyaluronic Acid	α-toc	Injectable grafted copolymer	[64]
PNIPAM-b-PCL-b-PNIPAM	α-toc	Nanoparticles	[65]
SA-LA	α-toc	Nanoencapsules	[66]

**Table 4 materials-14-03691-t004:** Method of Vitamin E detection when it is combined with biomaterials.

Technique	Combined Material	Molecule Detected	Method	Information	Ref.
**HPLC**	UHWMPE	α-tocopherol	HPLC connected to UV/Vis diode array detector at 297 nm, construction of calibration curve of HPLC peak area.	Quantitative	[108]
	UHWMPE	α-tocopherol	HPLC connected to UV/Vis diode array detector, construction of calibration curve of absorbance peak area at 290 nm	Quantitative	[109]
	Collagen mesh	α-tocopherol	HPLC connected to a fluorescence detector, detection at excitation wavelength of 290 nm and emission wavelength of 330 nm	Quantitative	[112]
	Alginate and hyaluronate film	α-tocopherol acetate	HPLC connected to UV/Vis diode array detector, construction of calibration curve of absorbance peak area at 285 nm	Quantitative	[44]
	Hyaluronic-acid-based β-cyclodextrin copolymer	α-tocopherol	HPLC connected to UV/Vis diode array detector	Quantitative	[64]
	PNIPAM-b-PCL-b-PNIPAM copolymer	α-tocopherol	HPLC equipped with a differential refraction index detector	Quantitative	[65]
**UV-VIS**	UHWMPE	α-tocopherol	Construction of calibration curve of absorbance peak area at 290 nm	Quantitative	[110]
	UHWMPE	α-tocopherol	Analysis of reflectance spectra which presents a minimum around 290 nm and a decrease of reflectance at 400–500 nm.	Detection	[111]
	Hyaluronic acid	α-tocopherol succinate	Construction of calibration curve of absorbance peak area at 285 nm	Quantitative	[116]
	PLA+PCL	α-tocopherol acetate	Construction of calibration curve of absorbance peak area at 284 nm	Quantitative	[42]
**Colorimetric Assay**	UHWMPE	α-tocopherol	The yellowing of the sample was analysed through three parameters (a,b,L) of CIELAB colour space, and a calibration curve of colour distances was constructed.	Quantitative	[111]
**FTIR-ATR**	UHWMPE	α-tocopherol	Analysis of peaks. For quantitative analysis, calibration curve of these peaks is needed.	Analysis of Vitamin E transformation products in polymer samples prior to extraction and quantitative.	[108]
	Collagen	α-tocopherol	Analysis of main peaks	Characterization of film	[45]
**FTIR-ATR**	Magnetite	α -tocopheryl succinate	Analysis of main peaks	Characterization of chemical modification of nanoparticles	[71]
	Chitosan	α-tocopherol	Analysis of peaks	Physical bonds and chemical interactions are reflected by changes in characteristic spectral peaks.	[115]
	Chitosan	α-tocopherol	Analysis of peaks	Characterization of nanoparticles	[59]
	PCL/PLA	α-tocopherol acetate	Analysis of peaks	Characterization of membranes	[42]
	Soluplus	α-tocopherol	Analysis of peaks	Analysis of bonding between Soluplus/vitamin E	[114]
	Polyethylene	α-tocopherol	Analysis of peaks from 600–4000 cm ^−1^	Analysis of interaction between vitamin E and chitosan	[113]
**XPS**	Polyethylene	α-tocopherol	All binding energies were referenced to the C1s peak at 285 eV.	Analysis of covalent bonding	[113]
**DPPH**	Polyethylene	α-tocopherol	The scavenging activity was estimated RSA (%) = (1 − (A sample/Acontrol)) × 100, measuring the adsorption at 515 nm after 30 min in dark condition.	Radical scavenging activity evaluation	[113]
	Chitosan	α-tocopherol	The scavenging activity was estimated RSA (%) = (1 − (A sample/Acontrol)) × 100, measuring the adsorption at 517 nm after 30 min in dark condition.	Radical scavenging activity evaluation	[115]
	Collagen/chitosan	α-tocopherol	DPPH were measured by the adsorption at 517 nm after 30 min in dark condition. DPPH loss which is a concentration of DPPH radicals reacted with antioxidants.	Antioxidant activity	[112]
**Contact Angle**	Polyethylene	α-tocopherol	Contact angle titrations were performed by measuring sets of contact angles at each pH value.	Analysis of hydrophobic behaviour as pH increases	[106]
	PLA	α-tocopherol	Static contact angle	Analysis of material wettability change	[48,49]

## Data Availability

Not applicable.

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
