# Peer review of "Vitamin E: A Review of Its Application and Methods of Detection When Combined with Implant Biomaterials"

_materials, 2021, doi:10.3390/ma14133691_

Round 1

Reviewer 1 Report

The review comprehensively describes the structure, properties and applications of tocopherols and tocotrienols. The applications in different biomedical fields are properly taken into account and detailed picture of the vitamin E specific properties and behaviour is well integrated to the respective fields. 

The last chapter of the review, devoted to the methods for vitamin E determination, would be very valuable from practical point of view. All Tables in the review are comprehensive and make easy the access t the information  collected in the review.

English, although generally correct, should be checked again.

Reviewer 2 Report

The manuscript "Vitamin E: a review of its application and methods of detection when combined with implant biomaterials" contains interesting data with could have significant use for many researchers.

As general remarks, the references seems to be old, should be updated with more items from the recent literature. Also, a major revision of English grammar and style is needed, since many paragraphs lose their meaning.

Below some points that must be improved:

1. Line 41: "and compared to other isoforms, has the highest blood concentration and for this, is the subject of most research". The highest concentration is not the main reason of the research interest in alpha Tocopherol; there are many other reasons. The authors should modify the statement.

2. Line 54: "α-Tocopherol is located within the phospholipid membrane of the cell". Is Vitamin E always in the membrane? The authors should add a brief explanation when this happens.

3. Paragraphs starting at line 60: Using many times "well known" for scientific information is not appropriate in a review.

4. Line 107: "Tocopherol is able to reduce biofilm formation capacity of a big range of strains (S.aereus and S. epidermidis), without any correlation to the Gram negative or positive group". The phrase is confusing and the authors have to reformulate it.

5. Line 130: "The success of this type of incorporation is guaranteed by the very good biocompatibility of vitamin E". "Incorporation" means the introduction of Vitamin E in blend with polymer, and cannot be guaranteed by biocompatibility! The sentence must be rewritten.

6. Line 145: "coatings are created by the adsorption of vitamin E on the metal surfaces by leaving the surfaces in the vitamin solution". Not only the surface, but the whole metallic sample is kept in the vitamin solution. The sentence must be rewritten.

7. Line 148: The paragraph starting with "Bidossi et al. made interesting studies on the…" refers to the antimicrobial activity of Vitamin E derivatives and it has nothing to do with the subsection of coatings for implants; should be moved to the section above.

8. Line 171: "by creating a wound dressing with other polymers that provide the physical structure…" seems that a part of phrase is missing. The authors must rewrite it to make it clear.

9. Line 173: The hydrogels and scaffolds are not structures, are pharmaceutical formulations or colloidal systems (in the case of hydrogels).

10. Line 238: The title of the section 2.2 should be changed, since contains applications of Vitamin E, both as active substance and as drug delivery system (as a suggestion "Pharmaceutical use of Vitamin E").

11. Line 261: The whole phrase is too long and became confusing. The authors should reformulate it in a more concise manner.

12. Line 277: "Another important biopolymer used in drug delivery is definitely hyaluronic acid, a hydrophilic gel". Hyaluronic acid is a substance (polymer), not a gel (it forms gel when is dissolved in solvents); the correction must be made.

13. Line 290: Why the authors discuss in many details the encapsulation of Vitamin E in polymeric drug delivery systems and they don’t even mention other recent research on formulations based on liposomes, microemulsions, etc? Some comments on other vector for the encapsulation of Vitamin E must be added.

14. Line 334: More details should be provided in order to support the conclusion regarding the optimization of the drug delivery system (otherwise it remains a contentless statement).

15. Line 341: The title of the section 3 should be modified since it is not only about "detection", it contains also analytical methods for the quantification of the Vitamin E.

16. Line 344: A reference should be added, in order to support the statement.

17. The content of the section 3 should be organized, while in this moment qualitative and quantitative methods are randomly mixed. The purpose of the methods (quantification or study of the interaction between Vitamin E and other materials) are also mixed.

18. Conclusion section must be expanded, in order to emphasize the main added value of the paper. Some sentences on future perspective should be added.

Reviewer 3 Report

This paper is a comprehensive review study on the role of Vitamin E for biomedical application. This paper is a sound study that summarizes different application of vitamins E fr biomedical purposes, including tissue engineering, coating and etc as well as different method to detect vitamins. 

This paper provides a thorough review of the recent papers published in this research field. However, the inclusion factor is not clear in the paper. The author should include a statement elaborating on the inclusion factor and the number of papers that were reviewed in this paper. Also, the authors should include literature studied biomechanical effects of Vitamin E, in particular for coating applications.

Round 2

Reviewer 2 Report

The authors performed most of the suggested modifications and improved the manuscript. Some minor revisions of English language (spelling and style) are still necessary (e.g. line 470 "ince" is probably "since").